# Comparison of Different Machine Learning Classifiers for Glaucoma Diagnosis Based on Spectralis OCT

**DOI:** 10.3390/diagnostics11091718

**Published:** 2021-09-19

**Authors:** Chao-Wei Wu, Hsiang-Li Shen, Chi-Jie Lu, Ssu-Han Chen, Hsin-Yi Chen

**Affiliations:** 1Graduate Institute of Medicine, College of Medicine, Kaohsiung Medical University, Kaohsiung City 807378, Taiwan; chaowei196@gmail.com; 2Department of Ophthalmology, Kaohsiung Medical University Hospital, Kaohsiung Medical University, Kaohsiung City 807378, Taiwan; 3Graduate Institute of Business Administration, Fu Jen Catholic University, New Taipei City 242062, Taiwan; shune@uch.edu.tw (H.-L.S.); 059099@mail.fju.edu.tw (C.-J.L.); 4Artificial Intelligence Development Center, Fu Jen Catholic University, New Taipei City 242062, Taiwan; 5Department of Information Management, Fu Jen Catholic University, New Taipei City 242062, Taiwan; 6Department of Industrial Engineering and Management, Ming Chi University of Technology, New Taipei City 243303, Taiwan; 7Center for Artificial Intelligence & Data Science, Ming Chi University of Technology, New Taipei City 243303, Taiwan; 8Department of Ophthalmology, Fu Jen Catholic University Hospital, New Taipei City 24352, Taiwan; 9School of Medicine, College of Medicine, Fu Jen Catholic University, New Taipei City 242062, Taiwan

**Keywords:** optical coherence tomography (OCT), machine learning, glaucoma

## Abstract

Early detection is important in glaucoma management. By using optical coherence tomography (OCT), the subtle structural changes caused by glaucoma can be detected. Though OCT provided abundant parameters for comprehensive information, clinicians may be confused once the results conflict. Machine learning classifiers (MLCs) are good tools for considering numerous parameters and generating reliable diagnoses in glaucoma practice. Here we aim to compare different MLCs based on Spectralis OCT parameters, including circumpapillary retinal nerve fiber layer (cRNFL) thickness, Bruch’s membrane opening-minimum rim width (BMO-MRW), Early Treatment Diabetes Retinopathy Study (ETDRS) macular thickness, and posterior pole asymmetry analysis (PPAA), in discriminating normal from glaucomatous eyes. Five MLCs were proposed, namely conditional inference trees (CIT), logistic model tree (LMT), C5.0 decision tree, random forest (RF), and extreme gradient boosting (XGBoost). Logistic regression (LGR) was used as a benchmark for comparison. RF was shown to be the best model. Ganglion cell layer measurements were the most important predictors in early glaucoma detection and cRNFL measurements were more important as the glaucoma severity increased. The global, temporal, inferior, superotemporal, and inferotemporal sites were relatively influential locations among all parameters. Clinicians should cautiously integrate the Spectralis OCT results into the entire clinical picture when diagnosing glaucoma.

## 1. Introduction

Glaucoma is a potentially blinding disease, characterized by progressive degeneration of the retinal ganglion cells, resulting in distinct changes of the optic nerve head and corresponding visual field defect [1]. By observing the structural changes of the retinal nerve fiber layer, neuroretinal rim, or inner layer of macula, it is possible to detect the potential glaucoma patients [2].

Optical coherence tomography (OCT) is a non-invasive technology that enables high-resolution cross-sectional images of ocular tissues and provides objective quantitative data that have good reproducibility [3,4]. Through the various scan protocols provided by OCT, the subtle structural change caused by glaucoma can be detected earlier, even before the visual field loss would appear [5].

Although the abundant OCT parameters can give the clinicians a comprehensive understanding of the ocular structural changes, they may also cause confusion especially when the classification results conflict, usually seen in early-stage glaucoma due to the subtle structural changes [6]. There have been many studies investigating the diagnostic accuracy of parameters between normal and glaucomatous eyes [7,8,9,10,11,12,13], but most of them focused on individual parameters. However, in actual clinical situations, clinicians usually make the final diagnosis after considering all the parameters.

Machine learning classifiers (MLCs) are well-established analytical methods especially good at detecting the relationship between a huge amount of input parameters, eventually facilitating the diagnosis of a condition [14]. In fact, some reports suggest that MLCs are as good as [15,16], or even better [17,18,19,20,21,22,23,24] than, currently available techniques for glaucoma diagnosis. However, what most people criticize MLCs for is the analysis process being a “black box” [25,26,27], for it produces results based solely on the input data using an algorithm, which prevents clinicians from understanding how variables are being combined to make such a prediction. Through program analysis methods developed in recent years, we can study and analyze the importance of the included parameters in specific models, and ophthalmologists may obtain clinical insight from the explanation [24,25,28,29,30]. In this study, we aim to build MLCs based on clinical OCT data provided by Heidelberg Spectralis spectral-domain OCT (SD-OCT) to evaluate the diagnostic accuracy of MLCs and the importance of OCT parameters in diagnosing glaucoma of varying severities.

## 2. Materials and Methods

### 2.1. Participants

This cross-sectional study included 470 eyes from 265 participants. The patients with glaucoma were recruited from a group of patients who had received at least 6 months of regular follow-up at the Glaucoma Service of the Department of Ophthalmology at Fu Jen Catholic University Hospital (FJUH) between April 2019 to December 2021. Subjects with normal eyes were recruited from the out-patient clinic and staff at the Fu Jen Catholic University Hospital during the study period. This research adhered to the tenets of the Declaration of Helsinki. Informed consent was obtained from each participant, and the study was approved by the Institutional Review Board of FJUH (FJUH109020 and FJUH109021).

All the study subjects underwent complete ophthalmic examination, including slit-lamp biomicroscopy, measurement of intraocular pressure (IOP), stereoscopic fundus examination, and standard automated perimetry. Visual fields were carried out on a Humphrey Visual Field Analyzer (model 750, Carl Zeiss Meditec, Dublin, CA, USA) with a standard white-on-white 30-2 field with the standard full threshold program. Demographic and clinical information was recorded, including age, gender, best-corrected visual acuity (BCVA), IOP, spherical equivalent (SE), mean deviation (MD), and pattern standard deviation (PSD). 

All participants were required to have a best-corrected visual acuity of 20/40 or better and a spherical equivalent within the −6.0 diopter (D) to +3.0 D range. Patients with significant ocular media opacity, coexisting retinal disease, or history of refractive or vitreoretinal surgery were excluded from this study. In addition, for increasing the imaging quality and accuracy, subjects with marked peripapillary atrophy were excluded to avoid instrumentation problems in the algorithms used to find the layers.

Normal control eyes had normal findings on clinical examination, IOPs lower than 21 mmHg, no history of increased IOP, no family history of glaucoma, normal-looking optic disc heads, and normal visual field results. A normal visual field result was defined as an MD and PSD within 95% confidence limits and a glaucoma hemifield test result within normal limits.

Eyes were defined as glaucomatous if there were glaucomatous optic neuropathy (GON) and corresponding visual field loss. We defined GON as either inter-eye cup-disc ratio asymmetry > 0.2, rim thinning or notching, peripapillary hemorrhages, or cup-disc ratio ≥ 0.2 [31,32]. Glaucomatous visual field defects were evaluated based on the following criteria: ≥2 contiguous points with a pattern deviation sensitivity loss of *p*-value < 0.01, ≥3 contiguous points with a sensitivity loss of *p*-value < 0.05 in the superior or inferior arcuate areas, or a 10-decibels (dB) difference across the nasal horizontal midline at ≥2 adjacent locations and an abnormal result on the glaucoma hemifield test [33]. Also, visual field defects had to be reproducible on at least one occasion. A visual field was considered reliable if there was <20% fixation loss, as well as <20% false-positive and false-negative rates. Both eyes of each participant were included if they were glaucomatous. All eyes with secondary glaucoma or evidence of visual field defects consistent with other diseases were excluded.

Using the Hodapp–Parrish classification [34], glaucomatous eyes were stratified into three groups for further analysis: early glaucoma (MD > −6 dB), moderate glaucoma (−12 dB < MD ≤ −6 dB), and severe glaucoma (MD ≤ −12 dB).

### 2.2. Spectralis OCT Imaging

All participants were examined with Spectralis (Heidelberg Engineering Inc., Heidelberg, Germany) SD-OCT, and the Anatomic Positioning System (APS) was used in the process. The APS creates an anatomic map using two fixed structural landmarks: the center of the fovea and the center of the Bruch’s membrane opening. All scan protocols are automatically oriented according to the patient’s anatomic map to obtain precise measurements of relevant structures and ensure accurate comparisons with reference data. Optic nerve head radial and circular (ONH-RC) scan protocol and the posterior pole horizontal (PPoleH) scan protocol were used in the current study to get four groups of parameters: circumpapillary retinal nerve fiber layer (cRNFL) thickness, Bruch’s membrane opening-minimum rim width (BMO-MRW), macular Early Treatment Diabetic Retinopathy (ETDRS) thickness map, and posterior pole asymmetry analysis (PPAA). The two scan protocols can be obtained in the new Glaucoma Premium Module Edition (GPME) software. All the above scans were performed by experienced technicians. No manual correction was applied to the auto-segmented OCT output. Images that had a quality index of less than 20 or images with artifacts were excluded.

The ONH-RC scan protocol consisted of three circular scans with diameters of 3.5 mm, 4.1 mm, and 4.7 mm to acquire the cRNFL thickness, and 24 equally spaced radial B-scans to get the BMO-MRW. Both of the circular and radial scans were centered on the optic nerve head. The BMO-MRW measurements were obtained by calculating the shortest distance between the BMO points and the internal limiting membrane, which was automatically identified by the build-in software. The cRFNL and BMO-MRW measurements were displayed in seven parts: global (G), temporal (T), superotemporal (TS), superonasal (NS), nasal(N), inferonasal (NI), and inferotemporal (TI) area. Each part averaged the corresponding measurements of the area. All of the seven BMO averages would be used to build the MLCs, but only the seven cRNFL averages generated by the 3.5 mm diameter circle would be included in the subsequent analysis.

The PPoleH scan protocol consists of 61 horizontal B-scans, centered on the fovea, oriented to the fovea-disc axis, and symmetrically distributed in the upper and lower hemispheres. It provides the automated segmented thickness map of each retinal layer, including macular retinal nerve fiber layer (mRNFL), ganglion cell layer (GCP), inner plexiform layer (IPL), inner nuclear layer, outer plexiform layer, outer nuclear layer, photoreceptors layer, and retinal pigment epithelium. These measurements are displayed in two modes: ETDRS mode and PPAA mode.

In the ETDRS mode, the thickness maps of total macular thickness and all eight retinal layers are provided in nine subfields as defined by the ETDRS. The diameters of inner, intermediate, and outer rings are 1, 3, and 6 mm separately. The average of all points with the inner ring was defined as central thickness (C). The intermediate ring was divided into four sectors designated as inner superior (S1), inner nasal (N1), inner inferior (I1), and inner temporal (T1) sector, and so was the outer ring, designated as outer superior (S2), outer nasal (N2), outer inferior (I2) and outer temporal (T2) sector. The values of the nine zones of total macular thickness, mRNFL, GCL, and IPL thickness were used in the following analysis.

In the PPAA mode, the thickness of the full retina and each retinal layer are measured, averaged and displayed in an 8 × 8 grid, corresponding to the central 24° area of the posterior pole. We only included the 64 measurements of full retinal thickness, labeled with the first number representing the order from top to bottom and the second number representing the order from temporal to nasal site.

In total, 114 parameters (7 RNFL, 7 MRW, 36 ETDRS, 64 PPAA) were used for subsequent MLC establishment (Appendix A). All the included subjects have no missing value in these 114 measurements.

### 2.3. Machine Learning Classifiers

Five machine learning algorithms, namely conditional inference trees (CIT), logistic model tree (LMT), C5.0 decision tree, random forest (RF), and extreme gradient boosting (XGBoost), are used to build the early, moderate, and severe glaucoma classification models. MLCs for normal and all glaucomatous eyes are also constructed. These machine learning algorithms have been widely applied in various healthcare and/or medical informatics applications and do not have a prior assumption about data distribution [35,36,37,38]. The multivariate logistic regression (LGR) was used as a benchmark for comparison. The 114 Spectralis OCT parameters are used as independent variables for all used methods to build glaucoma classification models.

The CIT uses recursive partitioning of dependent variables and embeds a permutation test to discriminate between significant and insignificant improvements [39]. The LMT combines decision tree and LGR where independent variables are splitting by a logistic variant information gain, LGR models at all nodes are produced by the LogitBoost, and its final decision tree is pruned by using the method of classification and regression trees [40]. The C5.0 decision tree is a successor of the C4.5 decision tree. The independent variables are splitting based on the entropy. The final tree is pruned by the binomial confidence limit method [41]. The RF is a bagging technique that builds a set of decision trees, that is, a forest, by randomly selecting samples and independent variables. The prediction class of RF is considered the majority vote [42]. The XGBoost is an extendible version of gradient boosting machines that was developed for the pursuit of pushing the limits of prediction performance and computational speed. In the boosting technique, boosted models are added recursively to adjust the residuals made by existing models. The final prediction is based on the majority votes [43].

In this study, all methods were implemented in R software version 3.6.2 (R core team, Vienna, Austria). For modeling LGR, CIT, LMT, C5.0, RF, and XGBoost models, the corresponding R packages “blorr” [44], “partykit” [45], “RWeka” [46], “C50” [47], “randomForest” [48], and “xgboost” [49] are used in this study. The R package “caret” [50] was used to tune the hyper-parameters of each machine learning method. 

In building the predictive model, we first reorganized the dataset into three sub-datasets, normal vs. early glaucoma, normal vs. moderate glaucoma, and normal vs. severe glaucoma. Next, we randomly divided each sub-dataset with a specific percentage several times. A split of 80–20 is used, where the training dataset gets 80% and the testing dataset gets 20% of the total data. 

During the training stage, we attempted to tune the hyper-parameters of each MLC to find out a model whose prediction performance is relatively good. We conducted the pipeline of a random search, 10-fold cross-validation, and a metric evaluation using “caret” R package. Within the pipeline, the random search generated 50 sets of hyper-parameters of each classifier that followed uniform distribution. For each hyper-parameter set, the training dataset was further divided into 10 equal-sized folds. Of the 10 folds, 9 folds are used as the real training dataset for training the classifier, whereas the remaining single fold is treated as the validation dataset for evaluating the performance by using the area under the receiver operating characteristic curve (AUC). The 10-fold cross-validation was repeated 10 times by changing the fold of the validation dataset and an average AUC of a given set of hyper-parameters was produced. The higher the average AUC is, the better the hyper-parameters of a classifier will be.

During the testing stage, the features were fed into the best model to get the predicted classification result of each sample. Finally, all predicted results were compared with their corresponding labels to generate a confusion matrix that can be used to calculate the performance metrics of a classifier.

For each classifier, the above process was randomly repeated 100 times for bootstrap samples of the data. The 100 recorded accuracy, sensitivity, specificity, and AUC of each classifier were averaged separately as the performance metric of each MLC. The mean rank aggregation method [30,51] was used to average the yielded 100 results of variable importance ranking of a classifier to produce a combined variable importance ranking for each MLC. 

### 2.4. Statistical Analysis

All statistical analyses were carried out using the statistical programming language R software of version 3.6.2 (R core team, Vienna, Austria). Demographics and clinical characteristics of the study groups were compared using the Mann–Whitney U test for quantitative variables and the Chi-square test for categorical variables.

## 3. Results

The demographics and clinical characteristics of the study groups are given in Table 1. The age of the glaucoma groups was significantly older than the normal group (*p* < 0.001), and the BCVA of the glaucoma groups was significantly worse than the normal group (*p* < 0.001). No significant difference was observed in the IOP and spherical equivalent. Visual field parameters including MD and PSD showed significant differences between the two groups (*p* < 0.001).

The LGR, CIT, LMT, C5.0, RF, and XGBoost methods were used to build classification models for discriminating all glaucomatous eyes from normal eyes. They were also used for distinguishing between early, moderate, and severe glaucoma from normal conditions. The model performance, that is, the mean and the standard deviation (SD) of accuracy, sensitivity, specificity, and AUC, of all models for all, early, moderate, and severe glaucoma datasets are respectively presented in Table 2, Table 3, Table 4 and Table 5. The receiver operating characteristic (ROC) curves as well as 95% confidence interval (CI) of mean AUCs of all classifiers for all, early, moderate, and severe glaucoma datasets are also demonstrated in Figure 1. 

It can be observed from Table 2 and Figure 1a that all five MLCs can generate better performance than the benchmark LGR method (mean AUC = 0.7788) in all the glaucoma eyes datasets. RF provides the highest value of average accuracy (0.8818), average sensitivity (0.9166), average specificity (0.8507), and average AUC (0.9459). RF is the best predictive model in all the glaucoma eyes datasets and its mean AUC and 95% CI is 0.9459 ± 0.0047.

Referring to Table 3 and Figure 1b, it can be found that that the model performance of MLCs outperforms that of the benchmark LGR method (mean AUC = 0.6860) in the early dataset. The model performance of CIT and LMT are lower than that of C5.0, RF, and XGBoost algorithms. XGBoost receives the highest value of average sensitivity (0.7500) and RF generates the highest values of average accuracy (0.8550), average specificity (0.9189), and average AUC (0.9073). RF is the best predictive model in the early glaucoma dataset and its mean AUC, and 95% CI is 0.9073 ± 0.0071.

According to Table 4 and Figure 1c, we can find that the model performance of MLCs also outperforms that of LGR in the moderate dataset and all of the MLCs had excellent diagnostic performance, each with a mean AUC value greater than 0.9. The model performance of CIT is lower than other MLCs. XGBoost receives the highest value of average sensitivity (0.8392) and RF generates the highest values of average accuracy (0.9327), average specificity (0.9646), and average AUC (0.9655). RF is still the best predictive model in the moderate dataset and its mean AUC, and 95% CI is 0.9655 ± 0.0058.

Table 5 and Figure 1d show that the model performance of MLCs also outperforms LGR in the severe dataset. There is not much difference in the model performance among MLCs. CIT receives the highest value of average sensitivity (0.8787) and RF generates the highest values of average accuracy (0.9536), average specificity (0.9797), and average AUC (0.9841). RF is still the best predictive model for the severe glaucoma dataset and the mean AUC, and 95% CI is 0.9841 ± 0.0044.

As the MLCs can generate promising classification performance for all three datasets and outperform the LGR model, they can be used to identify important variables for discriminating glaucomatous eyes from the normal eyes, and early, moderate, and severe glaucoma from normal conditions by ranking the importance of each variable within different classifiers. For each model, the most important variable was ranked first (i.e., 1). On the contrary, the variable with the lowest importance was ranked as the last (i.e., 114). Appendix A show the importance ranking of each variable of each classifier for all, early, moderate, and severe datasets, respectively. Taking Appendix A as an example, GCL_T2 is found to be the most important variable in the C5.0 model, followed by MRW_TI and cRNFL_TS. For the ranking result of the RF model, the most important area is also GCL_T2, but the second- and third-most important areas are cRNFL_G and GCL_T1, respectively. It can be seen that different classifiers can generate different variable importance ranking results at different severities of glaucoma.

In order to build a more robust variable importance ranking result by taking all the results from the MLCs into account, we average the rank value of each variable in all five MLCs. Table 6 lists the overall variable importance ranking of each variable for early, moderate, severe, and all glaucoma eyes. The averages were regarded as the mean importance of a single variable in the five MLCs. For distinguishing early glaucoma from normal eyes, the top five important predictors in order of importance are GCL_T2, GCL_T1, MRW_G, cRNFL_TS, and MRW_TI. For identifying moderate glaucoma from normal eyes, the top five important predictors in order of importance are GCL_T2, cRNFL_G, GCL_T1, MRW_G, and cRNFL_TS. For distinguishing severe glaucoma from normal eyes, the top five important predictors in order of importance are MRW_TI, cRNFL_G, GCL_I1, mRNFL_I2, and MRW_G. For discriminating all glaucoma eyes from normal eyes, the top five important predictors in order of importance are GCL_T2, GCL_T1, cRNFL_G, MRW_G, and MRW_TI.

## 4. Discussion

OCT has played an important role in glaucoma management, especially in early glaucoma detection. Though OCT parameters in diagnosing moderate and advanced glaucoma are of lower clinical importance, the role of OCT parameters in the evaluation of glaucoma progression is meaningful. Several studies worked on the analysis of OCT parameters in different severities of glaucoma. Mittal et al. surveyed which parameter of Cirrus and RTVue OCT has the highest ability to discriminate between early, moderate, and advanced glaucoma [52]. Ustaoglu et al. tried to determine the discriminating performance of the macular GC-IPL parameters between all the consecutive stages of glaucoma and to compare it with the discriminating performances of the RNFL and ONH parameters [53]. Chua et al. compared the diagnostic ability of macular intraretinal layer thickness with cRNFL thickness for detection of early, moderate, and advanced glaucoma [13]. Similarly, here, we tried to use machine learning methods to build reliable classifiers and study the importance of different OCT parameters in glaucoma of varying severities. 

RF was demonstrated as the best or better method in other studies with similar study designs. Barella et al. analyzed cRNFL and optic nerve head parameters in SD-OCT using 10 machine learning methods and reported that RF had the best diagnostic performance among the MLCs to discriminate between normal and early to moderate glaucoma eye with an AUC of 0.877 [16]. However, the AUC obtained with RF was not significantly different from the AUC obtained with the best single OCT parameter in that study. Yoshida et al. used the RF method to analyze cRNFL, mRNFL, and ganglion cell-inner plexiform layer (GC-IPL) parameters in SD-OCT and reported that the RF method significantly improved the diagnosis of glaucoma with an AUC of 0.985 compared with using a single SD-OCT measurement [54]. Kim et al. compared four MLCs built with clinical features, cRNFL, and visual field parameters for discriminating between healthy and glaucomatous eyes, which showed that the RF classifier is the best one with an AUC of 0.979 [55]. Seo et al. investigated the diagnostic accuracy of six MLCs using BMO-MRW, cRNFL, and cRNFL color codes from SD-OCT to discriminate between early normal tension glaucoma patients from glaucoma suspects. The RF classifier was the second-best performing MLC with an AUC of 0.947, and the deep neural network model was the best one with an AUC of 0.966 [23]. These studies suggest that RF is a powerful and reliable machine learning method.

Since MLCs only use algorithms to derive prediction results from known input data, even if the inferred results are accurate, it is difficult for clinicians to understand why MLCs can produce such results. Therefore, in this study, we used the function of the “caret” package in R language to further understand how each parameter affected the classification model and try to open the “black box”. Regarding how the program evaluates the importance of the parameter, in short, if the changes of the parameter values will affect the predicted result of the classifier more, then this parameter is considered to be more important in the model’s decision. Similar methods have been used to analyze the importance of parameters in MLCs in other studies [28,30,56].

For the importance of different parameters in our models, the GCL measurements (GCL_T2, GCL_T1, GCL_I1)) in the macular ETDRS thickness map had the greatest influence in the diagnosis of early glaucoma, and BMO-MRW and cRNFL thickness were less important. This result suggested that the structural changes of the GCL were important in distinguishing early glaucoma from normal eyes in our model. Though in a previous study by Pazos et al., compared with pRNFL parameters in Spectralis OCT, macular parameters such as mRNFL, IPL, and GCL showed a lower diagnostic capacity to discriminate between healthy subjects and early glaucoma patients when used individually [12], it should be noted that when considering all the OCT parameters concurrently to build the prediction model, it was not impossible that GCL measurements contributed more because of the interaction between the parameters. 

As the severity of glaucoma increased, GCL measurements (GCL_T2, GCL_T1, GCL_I1) still had a certain influence in distinguishing moderate glaucoma from normal eyes, and the ranking of cRNFL measurements (cRNFL_G, cRNFL_TS, cRNFL_TI) was significantly moved forward. This could be explained by the significant changes in cRNFL thickness compared to the normal eyes as glaucoma progressed to the moderate stage. 

In the identification of severe glaucoma eye in our models, BMO-MRW, cRNFL, and GCL measurements all had a certain degree of importance, which was probably because of the obvious changes in all structures at the advanced stage of glaucoma. It was worth noting that there seemed to be a trend where BMO-MRW affected the models more because it had more values that fall into the top 10 important variables (MRW_TI, MRW_G, MRW_T, MRW_TS). In terms of distinguishing between glaucomatous and normal eyes, the GCL, BMO-MRW, and cRNFL were obvious key parameters for classification.

Neither mFT nor rFT measurements entered the top 10 in terms of importance. A possible explanation is that the full thickness provided less information for diagnosing glaucoma.

For the importance of the location of parameters, the global, temporal, inferior, superotemporal, inferotemporal sites were relatively influential locations for most of the parameters. These findings were consistent with previous research showing that glaucoma usually affects the superior and inferior temporal nerve fiber [57]. 

In varying severities of glaucoma, G, TS, and TI sites were all important locations of cRNFL. The TS sector was more important in early glaucoma diagnosis, and the G site is more important in advanced glaucoma. Interestingly, for BMO-MRW parameters, the G site was the more important location in discriminating both early and moderate glaucoma from normal eyes. While in the advanced stage, the importance of the TI, T, and TS sectors increased, and the TI sector was the most important location in the advanced stage. This result was consistent with previous findings proposed by McCann et al. where mean global cRNFL and TI MRW-BMO were the best parameters among all in distinguishing glaucomatous (average MD: −8.77 dB) from normal eyes [9].

For macular parameters such as GCP, IPL, and mRNFL in the ETDRS thickness map, T and I quadrants were the most important quadrants. The T quadrant was more important in the early and moderate stage, and I quadrant was more important in the advanced stage. Our result is similar to the reports presented by Choi et al. that the inferior inner macular layers were more vulnerable in advanced glaucoma [58].

Although MLCs had been proven to be excellent in helping glaucoma diagnosis by using multiple SD-OCT parameters in many studies, to our knowledge, few studies addressed how MLCs made the decisions. Yoshida et al. used the RF method, analyzing cRNFL, mRNFL, and GC-IPL measurements concurrently and obtained better diagnostic performance than that from using any single SD-OCT measurement [54]. The superior and inferotemporal mRNFL and GC-IPL measurements significantly contributed to the RF classifier in the discrimination of early glaucoma eyes from normal eyes. In another study by Oh et al., the extreme gradient boosting method was shown to be the best one for the diagnosis of glaucoma among four methods using five features coming from clinical data, visual field tests, and cRNFL parameters [28]. The mean MD of the glaucoma group was −10.24 dB. The superior quadrant and inferior quadrant of cRNFL were reported to have the strongest influence on the proposed XGboost model. Our results were similar to these two studies where GCL measurements were the most contributing variables in early glaucoma detection and the TS as well as TI sectors were the more important variables among the cRNFL parameters in detecting moderate to severe glaucoma.

There are some strengths to our study. Our research proved that MLCs can be utilized to identify glaucoma patients using OCT data with high accuracy. What is more valuable in this study is that we used MLCs to simulate how the clinicians determine which parameters are important and should be paid attention to when faced with such abundant parameters as in Spectralis OCT. By observing the ranking changes of parameters in different glaucoma stages, we may also better understand the pathophysiological mechanism of glaucoma. Moreover, papers using OCT data from Asian populations for machine learning to detect glaucoma patients are still limited. The emergence of our study can increase the diversities of relevant research and is helpful for further cross-ethnicity comparison and analysis in the future [59].

Though the results are meaningful, the present study had some limitations. First, age was significantly different between glaucoma and control groups. As age was known to be associated with RNFL loss [60], this may generate some bias in our study. Second, our study population was not large enough, which might not represent the whole population. A further larger sample is needed. Third, our research was based on the Taiwan Chinese population, and the current results might not be able to apply to other ethnic groups. Finally, the machine learning method generates models according to the input parameters of Spectralis OCT. However, when using different OCT machines with different designs or different scan protocols, the generated model is definitely different. Our models may also not be able to generate accurate predictions when using data from subjects outside our inclusion criteria. Further study by comparing different OCT machines is mandatory for further exploring this important issue in the future. 

## 5. Conclusions

In conclusion, to our knowledge, this is one of the few studies that combine RNFL, neuroretinal rim, and macular parameters concurrently to build reliable MLCs for glaucoma diagnosis and to analyze the importance of the included parameters. Although the results are meaningful, clinicians should cautiously integrate the Spectralis OCT results into the entire clinical picture when diagnosing glaucoma.

## Figures and Tables

**Figure 1 diagnostics-11-01718-f001:**
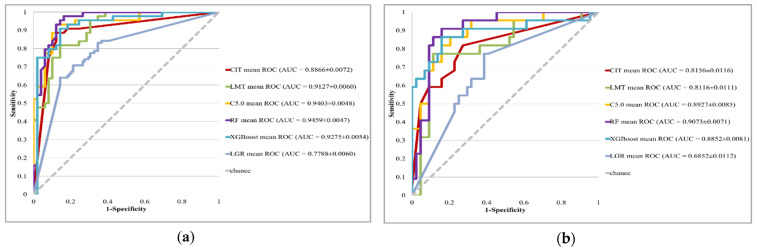
ROC curves of all classifiers for (**a**) all, (**b**) early, (**c**) moderate, and (**d**) severe glaucoma eyes.

**Table 1 diagnostics-11-01718-t001:** Demographics and clinical characteristics of the study groups.

Characteristics	Normal Eye (*n* = 224)	Glaucomatous Eye (*n* = 246)					
	Mean ± SD	Mean ± SD			*p* *		
Persons	114	151			-		
Age (year)	53.9 ± 16.0	61.2 ± 12.3			**<0.001**		
Gender (male/female)	41/73	83/68			**0.003**		
BCVA	0.96 ± 0.09	0.91 ± 0.11			**<0.001**		
IOP	14.1 ± 3.3	14.5 ± 2.6			0.230		
SE (D)	−1.13 ± 2.17	−1.10 ± 2.16			0.801		
		Early Stage (*n* = 114)		Moderate Stage (*n* = 61)		Severe Stage (*n* = 71)	
		Mean ± SD	*p* *	Mean ± SD	*p* *	Mean ± SD	*p* *
BCVA	0.96 ± 0.09	0.91 ± 0.11	**<0.001**	0.90 ± 0.11	**<0.001**	0.91 ± 0.11	**<0.001**
IOP	14.1 ± 3.3	14.8 ± 2.6	0.066	13.8 ± 2.6	0.405	14.5 ± 2.5	0.279
SE (D)	−1.13 ± 2.17	−1.17 ± 2.23	0.787	−1.16 ± 2.32	0.697	−0.72 ± 2.52	0.748
MD (dB)	−0.82 ± 1.32	−3.79 ± 1.10	**<0.001**	−8.31 ± 1.61	**<0.001**	−19.01 ± 5.49	**<0.001**
PSD (dB)	2.20 ± 1.29	4.38 ± 2.08	**<0.001**	8.87 ± 3.43	**<0.001**	11.40 ± 2.97	**<0.001**

BCVA: best-corrected visual acuity; IOP: intraocular pressure; SE: spherical equivalent; D: diopters; MD: mean deviation; PSD: pattern standard deviation. *p* *: level of statistical significance in comparison between groups using the Mann–Whitney U test (except for eye, gender, Chi-square). Bold text indicates statistically significant results (*p* < 0.05).

**Table 2 diagnostics-11-01718-t002:** Model performance of all methods for all glaucoma eyes.

Methods	Accuracy Mean (SD)	Sensitivity Mean (SD)	Specificity Mean (SD)	AUC Mean (SD)
LGR	0.7418 (0.03)	0.7479 (0.05)	0.7363 (0.05)	0.7788 (0.03)
CIT	0.8208 (0.04)	0.8507 (0.07)	0.7939 (0.07)	0.8866 (0.04)
LMT	0.8491 (0.04)	0.8825 (0.05)	0.8192 (0.06)	0.9127 (0.03)
C5	0.8709 (0.03)	0.8979 (0.05)	0.8468 (0.06)	0.9403 (0.02)
RF	0.8818 (0.03)	0.9166 (0.04)	0.8507 (0.06)	0.9459 (0.02)
XGBOOST	0.8639 (0.03)	0.8982 (0.05)	0.8331 (0.06)	0.9275 (0.03)

LGR: logistic regression; CIT: conditional inference tree; LMT: logistic model tree; RF: random forest; XGBOOST: extreme gradient boosting; AUC: area under the receiver operating characteristic (roc) curve.

**Table 3 diagnostics-11-01718-t003:** Model performance of all methods for early glaucoma eyes.

Methods	Accuracy Mean (SD)	Sensitivity Mean (SD)	Specificity Mean (SD)	AUC Mean (SD)
LGR	0.6916 (0.05)	0.5859 (0.11)	0.7442 (0.07)	0.6860 (0.06)
CIT	0.7729 (0.05)	0.7286 (0.10)	0.7950 (0.08)	0.8156 (0.06)
LMT	0.8006 (0.05)	0.6990 (0.09)	0.8514 (0.06)	0.8116 (0.06)
C5	0.8408 (0.04)	0.7041 (0.09)	0.9091 (0.05)	0.8927 (0.04)
RF	0.8550 (0.04)	0.7272 (0.08)	0.9189 (0.04)	0.9073 (0.04)
XGBOOST	0.8433 (0.04)	0.7500 (0.10)	0.8901 (0.05)	0.8852 (0.04)

LGR: logistic regression; CIT: conditional inference tree; LMT: logistic model tree; RF: random forest; XGBOOST: extreme gradient boosting; AUC: area under the receiver operating characteristic (roc) curve.

**Table 4 diagnostics-11-01718-t004:** Model performance of all methods for moderate glaucoma eyes.

Methods	Accuracy Mean (SD)	Sensitivity Mean (SD)	Specificity Mean (SD)	AUC Mean (SD)
LGR	0.8287 (0.05)	0.6625 (0.12)	0.8748 (0.06)	0.8187 (0.07)
CIT	0.8861 (0.05)	0.8175 (0.11)	0.9047 (0.06)	0.9035 (0.05)
LMT	0.9225 (0.04)	0.8008 (0.13)	0.9557 (0.04)	0.9553 (0.04)
C5	0.9200 (0.03)	0.7809 (0.12)	0.9580 (0.03)	0.9529 (0.04)
RF	0.9327 (0.03)	0.8159 (0.12)	0.9646 (0.03)	0.9655 (0.03)
XGBOOST	0.9293 (0.03)	0.8392 (0.12)	0.9540 (0.03)	0.9549 (0.04)

LGR: logistic regression; CIT: conditional inference trees; LMT: logistic model tree); RF: random forest; XGBOOST: extreme gradient boosting; AUC: area under the receiver operating characteristic (roc) curve.

**Table 5 diagnostics-11-01718-t005:** Model performance of all methods for severe glaucoma eyes.

Methods	Accuracy Mean (SD)	Sensitivity Mean (SD)	Specificity Mean (SD)	AUC Mean (SD)
LGR	0.8714 (0.04)	0.7700 (0.11)	0.9031 (0.05)	**0.9017** (0.05)
CIT	0.9219 (0.04)	0.8787 (0.09)	0.9358 (0.05)	**0.9416** (0.05)
LMT	0.9367 (0.03)	0.8580 (0.09)	0.9619 (0.03)	**0.9592** (0.04)
C5	0.9516 (0.03)	0.8687 (0.09)	0.9779 (0.03)	**0.9757** (0.03)
RF	0.9536 (0.02)	0.8737 (0.08)	0.9791 (0.02)	**0.9841** (0.02)
XGBOOST	0.9393 (0.03)	0.8651 (0.09)	0.9630 (0.03)	**0.9742** (0.03)

LGR: logistic regression; CIT: conditional inference trees; LMT: logistic model tree); RF: random forest; XGBOOST: extreme gradient boosting; AUC: area under the receiver operating characteristic (roc) curve.

**Table 6 diagnostics-11-01718-t006:** Overall variable importance ranking for early, moderate, severe, and all glaucoma eyes (only the first 10 important variables shown).

	Early		Moderate		Severe		All	
Rank	Variable Name	Average Rank	Variable Name	Average Rank	Variable Name	Average Rank	Variable Name	Average Rank
1	GCL_T2	1.0	GCL_T2	1.8	MRW_TI	2.0	GCL_T2	1.8
2	GCL_T1	3.4	cRNFL_G	2.0	cRNFL_G	3.0	GCL_T1	3.6
3	MRW_G	4.6	GCL_T1	3.0	GCL_I1	3.6	cRNFL_G	3.8
4	cRNFL_TS	4.8	MRW_G	5.6	mRNFL_I2	4.6	MRW_G	5.2
5	MRW_TI	5.6	cRNFL_TS	5.8	MRW_G	5.8	MRW_TI	5.2
6	cRNFL_G	5.8	GCL_I1	7.2	cRNFL_TI	8.0	GCL_I1	5.6
7	GCL_I1	7.4	cRNFL_TI	8.0	MRW_T	8.6	cRNFL_TS	5.6
8	IPL_T1	10.2	IPL_T2	9.8	MRW_TS	10.0	cRNFL_TI	10.6
9	MRW_N	12.8	mRNFL_I2	9.8	GCL_T1	10.6	MRW_NI	13.4
10	MRW_TS	13.0	MRW_N	11.8	cRNFL_TS	11.4	MRW_N	13.6

cRNFL: circumpapillary retinal nerve fiber layer; BMO-MRW: Bruch’s membrane opening opening-minimal rim width; mRNFL: macular retinal nerve fiber layer; GCL: ganglion cell layer; IPL: inner plexiform layer; T: temporal; TI: inferotemporal; N: nasal; TS: superotemporal; G: global; I: inferior. Note: These variables are colored according to the parameters to facilitate understanding of their changes (red: EDTRS macular thickness map, green: cRNFL, blue: BMO-MRW).

## Data Availability

Not applicable.

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
