# Peer review of "Comparison of Different Machine Learning Classifiers for Glaucoma Diagnosis Based on Spectralis OCT"

_diagnostics, 2021, doi:10.3390/diagnostics11091718_

Round 1

Reviewer 1 Report

The authors measured the diagnostic accuracy of MLCs and investigated the importance of OCT parameters by using SD-OCT for control and glaucoma patients (early, moderate, and severe group). It is necessary to explain the major and minor concerns as follows.

Major concerns)

  1. Mean values of visual acuity, intraocular pressure, and spherical equivalents in the control group, all patients with glaucoma, and in each group classified according to severity should also be presented. This is important data for understanding the basic information of patients.
  2. In clinical situations, ophthalmologists use OCT parameters to determine whether or not glaucoma is present, and then decide on a treatment strategy according to the disease severity. Before analysis according to the severity of glaucoma, the ROC curve among multiple MLCs of all glaucoma patients should be added, and variable importance ranking using five machine learning classifiers should also be investigated in the total glaucoma patients.
  3. It would be helpful to collect cases representative of each clinical early, moderate, and severe glaucoma patient and show which parameters were important.
  4. In Introduction, the authors will do an explainable program analysis rather than a black box, and how to explain this part should be explained in one paragraph in the discussion.
  5. Actually, studies on which parameters among OCT parameters are helpful for early detection of glaucoma are meaningful and have clinical significance. However, studies on which OCT parameters are helpful in diagnosing glaucoma in moderate and severe glaucoma are of low clinical importance. In moderate and severe glaucoma patients, the change in RNFL thickness in OCT is due to the floor effect, so it is important to judge the progression of glaucoma. However, it is questionable whether it is clinically meaningful to analyze the importance of OCT parameters by dividing groups according to severity in diagnosing glaucoma. Authors should describe this content using references.

Minor concerns)

  1. Lime 27: Does MLDs mean MLCs?

Author Response

Response to the Reviewer one’s comment

The authors measured the diagnostic accuracy of MLCs and investigated the importance of OCT parameters by using SD-OCT for control and glaucoma patients (early, moderate, and severe group). It is necessary to explain the major and minor concerns as follows.

  • Thanks so much for your support for our study. We have tried our best to address each of your concern and suggestion and have revised our work as your precious opinions.

Major concerns)

  1. Mean values of visual acuity, intraocular pressure, and spherical equivalents in the control group, all patients with glaucoma, and in each group classified according to severity should also be presented. This is important data for understanding the basic information of patients.

àReply: We have added the visual acuity, intraocular pressure, and spherical equivalent data in the Table 2 as your suggestion. (please see revised Table 2)

  1. In clinical situations, ophthalmologists use OCT parameters to determine whether or not glaucoma is present, and then decide on a treatment strategy according to the disease severity. Before analysis according to the severity of glaucoma, the ROC curve among multiple MLCs of all glaucoma patients should be added, and variable importance ranking using five machine learning classifiers should also be investigated in the total glaucoma patients.

àReply: Thanks for your valuable suggestion. We have added the results and corresponding table (Table 3) and figure (Figure 1) of using the MLCs for all glaucoma patients in the revised manuscript (Line 255-259). The variable importance ranking results and corresponding tables (Tables 7 and 11) of using the MLCS for all glaucoma patients are also added and discussed in the revised manuscript (Line 338-351, 377-390, 455-457).

  1. It would be helpful to collect cases representative of each clinical early, moderate, and severe glaucoma patient and show which parameters were important.

àReply: Thanks so much for your opinion. Because the study results already have 11 Tables sand 4 Figures, we think it is not really necessary to put OCT result each of early, moderate and severe glaucoma eye in the text.

  1. In Introduction, the authors will do an explainable program analysis rather than a black box, and how to explain this part should be explained in one paragraph in the discussion.

àReply: We had revised this part in Line 65-70 to make the “black box” concept much clearer and also added a paragraph in the discussion section from Line 424-432 to address how we opened the “black box” and propose other references using similar method.

  1. Actually, studies on which parameters among OCT parameters are helpful for early detection of glaucoma are meaningful and have clinical significance. However, studies on which OCT parameters are helpful in diagnosing glaucoma in moderate and severe glaucoma are of low clinical importance. In moderate and severe glaucoma patients, the change in RNFL thickness in OCT is due to the floor effect, so it is important to judge the progression of glaucoma. However, it is questionable whether it is clinically meaningful to analyze the importance of OCT parameters by dividing groups according to severity in diagnosing glaucoma. Authors should describe this content using references.

àReply: Thanks so much for your precious opinion. We have added three important references to clarify the meaning of OCT in managing different stage of glaucoma and we also added some important information in the discussion part. ( Line 394-406, New added Ref 52, 53 )

Minor concerns)

  1. Lime 27: Does MLDs mean MLCs?

àReply: Thanks so much your reminding. We have corrected this typo  error.

Reviewer 2 Report

The authors submitted a study about a very interesting topic such as the application of artificial intelligence to glaucoma patients.

The study is well written and conducted with good methodology but there is something missing about the overall meaning of this manuscript.

Which is the final goal of the manuscript? To build a glaucoma classification according to MLC using OCT data? Why? Which are the advantages for physicians?

According to data provided, the model elaborated is able to replicate the “human” diagnosis using visual field and clinical data. Moreover, in this study some kind of eyes, such as the ones with optic disc atrophy, have been excluded because of possible bias. Authors would need to clarify that any MLC is not able to include every kind of eyes.

The conclusion needs to be more strength, to provide more useful information for physicians.

The authors worked on many parameters but, in the end, this study has no utilities.

Author Response

Response to the Reviewers Two’s comments:

The authors submitted a study about a very interesting topic such as the application of artificial intelligence to glaucoma patients.

The study is well written and conducted with good methodology but there is something missing about the overall meaning of this manuscript.

  • Thanks so much for your support for our study. We have tried our best to address each of your concern. We hope our revised work could be better.

Which is the final goal of the manuscript? To build a glaucoma classification according to MLC using OCT data? Why? Which are the advantages for physicians?

àReply: Thank you for your reminding. Our goal of the manuscript is to build MLCs based on clinical OCT data provided by OCT to evaluate the diagnostic accuracy of MLCs and the importance of OCT parameters in diagnosing glaucoma of varying severities, which had been addressed in Line 54-60. The advantages of our study results are to help clinicians in using OCT results in glaucoma management. We have also added some strengths of our study in the discussion part. (please see Line 493-502)

According to data provided, the model elaborated is able to replicate the “human” diagnosis using visual field and clinical data. Moreover, in this study some kind of eyes, such as the ones with optic disc atrophy, have been excluded because of possible bias. Authors would need to clarify that any MLC is not able to include every kind of eyes.

àReply: Thanks so much for your concern. We have added some information about this limitation in the discussion part. (see Line 510-513)

The conclusion needs to be more strength, to provide more useful information for physicians.

àReply: Thank you so much for your suggestion. We had added some important findings in the conclusion. (please see Line 518-523)

The authors worked on many parameters but, in the end, this study has no utilities.

àReply: Thanks so much for your concern and opinion. The value of our study is to help clinicians in managing glaucoma by using OCT information with use. As far as we know that OCT could provide several parameters in optic nerve head area and macular areas, clinicians usually get confused about the enormous information that OCT report could provide. Our research proved that MLCs can be used to identify glaucoma patients using OCT data with high accuracy. We had added a paragraph from Line 493 to Line 502 to further address the clinical value of our study.

Reviewer 3 Report

Although the current paper is of interest for the field of machine learning and the diagnosis of glaucoma using Optical coherence tomography, the below concerns needs to be addressed:

The plagiarism check did not reveal a significant overlap with previously published data.

Lines 54-56, 63, etc..: Many key sentences lack references. The authors are encouraged to add adequate reference(s).

Line 168: 2.3The authors have joined the ML and statistical analysis in one paragraph. This is not highly advised. The authors are encouraged to add the statistical analysis section as separate paragraphs.

The statistical section is underdeveloped, and some critical information is lacking:

  1. The authors included 470 eyes from 265 participants. How did they choose this number? Did they try to carry out any sampling/experimental design before conducting the study? Determining the optimal sample size in the statistical analysis study could provide readers with the adequate number of participants needed to detect significant robust results! The GPower tool can do this calculation.

  1. student t-test was used to compare the quantitative data. The student t-test is not appropriate for the authors' data for the following reasons:
  2. the authors did not study the distribution of their investigated variables.
  3. Even if these variables show a normal distribution, no solid conclusions can be drawn because of the limited sample size. Thus it is highly recommended to use a non-parametric test such as the Mann-Whitney U test.
  4. for the logistic regression models, the authors did not specify it the model used was multivariate or univariate? What were the independent variables? This needs to be clarified.

Author Response

Response to the Reviewer Three’s comments:

Although the current paper is of interest for the field of machine learning and the diagnosis of glaucoma using Optical coherence tomography, the below concerns needs to be addressed:

The plagiarism check did not reveal a significant overlap with previously published data.

  • Thanks so much for your encouragement and support for our study.  

 Lines 54-56, 63, etc..: Many key sentences lack references. The authors are encouraged to add adequate reference(s).

àReply: Thanks so much for your kind reminding. We have added relevant references (Reference 6 for Line 54-56, Reference 14 for Line 63) to strengthen our viewpoint, and slightly revised the words and sentences in Lines 55-58 to make it clearer.

Line 168: 2.3The authors have joined the ML and statistical analysis in one paragraph. This is not highly advised. The authors are encouraged to add the statistical analysis section as separate paragraphs.

àReply: Thanks for your suggestion. We have separated ML and statistical analysis into 2 independent paragraphs. Statistical analysis section was presented in a single paragraph ( see Line 230- 235).

The statistical section is underdeveloped, and some critical information is lacking:

  1. The authors included 470 eyes from 265 participants. How did they choose this number? Did they try to carry out any sampling/experimental design before conducting the study? Determining the optimal sample size in the statistical analysis study could provide readers with the adequate number of participants needed to detect significant robust results! The G power tool can do this calculation.

àReply: Thanks so much for your question. Our study was case-control design but not sampling/ experiment design. Inclusion and exclusion criteria should be based on several clinical information, including refraction error, glaucoma severity from visual field data and imaging quality. Therefore, G power tool is not appropriate in our study.   

  1. student t-test was used to compare the quantitative data. The student t-test is not appropriate for the authors' data for the following reasons:

(1) the authors did not study the distribution of their investigated variables.

(2) Even if these variables show a normal distribution, no solid conclusions can be drawn because of the limited sample size. Thus, it is highly recommended to use a non-parametric test such as the Mann-Whitney U test.

àReply: We have revised the Table 2 by using Mann-Whitney U test for demographics and clinical characteristics of the study groups. The results of the statistical analysis are not much different from the previous one and do not affect the findings of our research.

  1. for the logistic regression models, the authors did not specify it the model used was multivariate or univariate? What were the independent variables? This needs to be clarified.

àReply: Thanks for your valuable suggestion. A multivariate logistic regression model is used in this study. The independent variables for all used methods including logistic regression model and the five MLCs are 114 Spectralis OCT parameters. This point was incorporated into the revised manuscript (Line 178-180).

Round 2

Reviewer 1 Report

The authors made modifications as suggested by the reviewers.

Author Response

The authors made modifications as suggested by the reviewers.

English language and style are fine/minor spell check required

-->Thank you so much for your review and support for our work.  Regarding your suggestion, we have corrected all the spelling of the words in the whole text.

Reviewer 2 Report

The authors improved the overall quality of this submission but different flwas previously reported still remain.

Author Response

Response to the Reviewer two’s comment

The authors improved the overall quality of this submission but different flwas previously reported still remain.

-->Thank you so much for your review and support for our work. The academic editor suggested us to provide one supplementary file for Table 1 and Table 7-10 to help the readers more clear of the main text of our study. He also suggested us to rewrite the conclusions part. We have tried our best to improve my work; and we hope the revised version could be better.

Reviewer 3 Report

The authors have answered my comments 

Author Response

The authors have answered my comments.

-->Thank you so much for your review and support for our work.